

# Teenage sleep and technology engagement across the week

Amy Orben[1,2] and Andrew K. Przybylski[3,4]

[1] Emmanuel College, University of Cambridge, Cambridge, UK
[2] MRC Cognition and Brain Sciences Unit, University of Cambridge, Cambridge, UK
[3] Department of Experimental Psychology, University of Oxford, Oxford, UK
[4] Oxford Internet Institute, University of Oxford, Oxford, UK

Corresponding author
Amy Orben, aco35@cam.ac.uk

## ABSTRACT

**Background:** Throughout the developed world, adolescents are growing up with increased access to and engagement with a range of screen-based technologies, allowing them to encounter ideas and people on a global scale from the intimacy of their bedroom. The concerns about digital technologies negatively influencing sleep are therefore especially noteworthy, as sleep has been proven to greatly affect both cognitive and emotional well-being. The associations between digital engagement and adolescent sleep should therefore be carefully investigated in research adhering to the highest methodological standards. This understood, studies published to date have not often done so and have instead focused mainly on data derived from general retrospective self-report questionnaires. The value of this work has been called into question by recent research showing that retrospective questionnaires might fail to accurately measure these variables of interest. Novel and diverse approaches to measurement are therefore necessary for academic study to progress.

**Methods:** This study analyses data from 11,884 adolescents included in the UK Millennium Cohort Study to examine the association between digital engagement and adolescent sleep, comparing the relative effects of retrospective self-report vs. time-use diary measures of technology use. By doing so, it provides an empirical lens to understand the effects of digital engagement both throughout the day and before bedtime and adds nuance to a research area primarily relying on retrospective self-report.

**Results:** The study finds that there is a small negative association relating digital engagement to adolescent sleep both on weekdays and weekend days (median standardized association $\beta_{weekday} = -0.06$ and $\beta_{weekend} = -0.03$). There is a more negative association between digital engagement and total sleep time on weekdays compared to weekend days (median standardized $\beta_{weekday} = -0.08$, median standardized $\beta_{weekend} = -0.02$), while there is no such difference when examining adolescents' bedtime. Surprisingly, and contrary to our expectations, digital technology use before bedtime is not substantively associated with the amount of sleep and the tardiness of bedtime in adolescents.

**Conclusions:** Results derived from the use of transparent Specification Curve Analysis methods show that the negative associations in evidence are mainly driven by retrospective technology use measures and measures of total time spent on digital devices during the day. The effects are overall very small: for example, an additional hour of digital screen time per day was only related to a 9 min decrease in total time spent sleeping on weekdays and a 3 min decrease on weekends. Using

digital screens 30 min before bed led to a 1 min decrease in total time spent sleeping on weekdays and weekends. The study shows that more work should be done examining how to measure digital screen time before interventions are designed.

# INTRODUCTION

British children now spend an average of 2 h and 11 min each day online, using a diverse range of smartphone, tablet and computer screens (*Ofcom, 2019*). This engagement oftentimes continues past bedtime: research shows that nearly two thirds (62%) of children aged 12–15 are granted permission by their caretakers to take their mobile phone to bed (*Ofcom, 2019*). Given this emerging norm, it is an understandable concern that digital screen engagement might impact both sleep quantity and quality (*Owens, 2014*). Yet it is not clear whether this common-sense perception is rooted in scientific evidence. Approximately two thirds of British children believe that they have "a good balance between screen time and doing other things," and therefore seem to think that they are not adversely affected by the ever-increasing presence of digital screens in their lives (*Ofcom, 2019*). This apparent divergence in views raises the question of whether commonly held apprehensions about digital screen engagement are empirically justified and whether the persistent use of digital screens really impairs sleep habits.

This larger question is growing into an increasingly pressing concern for both scientists and policymakers in various developed countries (*Davies et al., 2019*; *Viner, Davie & Firth, 2019*). The notion that digital screens have a consistent and negative effect on sleep is indeed worrying, especially as sleep has been identified as a key factor in adolescent development (*Owens, 2014*; *Scott, Biello & Woods, 2019*; *Woods & Scott, 2016*), impacting both executive function and emotional stability (*Beebe, 2011*; *Owens, 2014*). A loss of sleep resulting from digital device use should therefore be a topic of conversation for academics, parents, and policy makers alike, as they are stakeholders who would need to collaborate closely and rapidly to find solutions to this potential problem (*Woods & Scott, 2016*).

There are many mechanisms by which digital technology engagement could forestall sleep onset and drive decreases in adolescent sleep more broadly. Device use may simply displace time that could be spent sleeping—one cannot be sleeping and simultaneously scrolling through Instagram—though the directionality of this association is still unclear (*Levenson et al., 2017*; *Owens, 2014*; *Przybylski, 2018*). Digital engagement might facilitate a cascade of behaviors such as socializing which replace time sleeping (*Scott, Biello & Cleland, 2018*), or use might simply be physiologically stimulating to the extent it delays sleep onset (*Cain & Gradisar, 2010*), or elicits subjective emotional experiences like fear of missing out which provide tantalizing alternatives to sleep (*Scott, Biello & Cleland, 2018*). A growing body of correlational and experimental research has linked digital screen use

before bed to negative effects (*Harbard et al., 2016*; *Levenson et al., 2017*; *Orzech et al., 2016*) and has postulated that this could be due to later bed times (*Cain & Gradisar, 2010*; *Orzech et al., 2016*) or decreased relaxation (*Harbard et al., 2016*). Taken together, there are multiple plausible mechanisms linking digital screen use, decreasing sleep and later bedtimes, but little clarity about whether these exist and, if they do, which theoretical explanation gives the most accurate representation of ultimate causal links.

A possible reason for the lack of consistent and substantial evidence for linkages between digital engagement and adolescent sleep is the low quality of measurement available in current data sources. The extant literature examining adolescents' use of digital technologies relies, in many cases, on operationalizations of "screen time": a somewhat nebulous and questionable construct (*Orben, Etchells & Przybylski, 2018*; *Orben & Przybylski, 2019a*). The challenges associated with using "screen time" as a predictive variable is exacerbated by the use of retrospective self-report measures which require adolescents to estimate and report how much digital technology they engage with. Human participants in research are inherently bad at judging how much time they spend on a wide variety of activities, and "screen time" is no exception (*Boase & Ling, 2013*; *Schwarz & Oyserman, 2001*). An expanding body of research has made it abundantly clear that asking a teenager to estimate the amount of time they have allocated to digital screens is not an optimal measurement strategy (*Robinson, 1985*; *Scharkow, 2016*). Furthermore, inaccuracies in reporting are influenced by the intensity of a participants' screen time, creating a systematic bias which is difficult to adjust for statistically (*Vanden Abeele, Beullens & Roe, 2013*; *Wonneberger & Irazoqui, 2017*). Improved, diverse and convergent measurement approaches to gauge digital engagement are therefore necessary to ultimately obtain robust and replicable insights into the effect of this activity on adolescents and their sleep.

One promising alternative to retrospective self-report measures are time-use diaries: a measurement method where participants keep a structured journal about what they are doing throughout one or two study days. Broadly speaking, time-use diaries are a form of experience sampling methodology: a research approach which has been used to probe a wide range of analog pursuits including well-being, motivation, and relationship processes (*Reis, Gable & Maniaci, 2013*) and, more recently, digital technology use (*Masur, 2019*; *Verduyn et al., 2015*). More specifically, time-use diaries require adolescents to fill out what they were doing on a fixed interval or task-contingent basis, and provide a record of what kinds of activities a young person engages with during any given time window of their day (*Ipsos MORI, 2016*). If adolescents report engaging in digital activities, the diaries can be used to calculate whether participants used digital devices at all, how much time they spent using digital devices and when in the day these devices were used.

The empirical value of time-use diary measurements has been increasingly acknowledged by social science researchers who analyze and collect such data. These approaches have now been included in high-quality large cohort studies such as Growing Up in Ireland and the US Panel Study of Income Dynamics (*Williams et al., 2009*). Although this method falls short of the "perfect world" of direct behavioral tracking (*Andrews et al., 2015*; *David, Roberts & Christenson, 2018*), it represents a clear

diversification of current practices and is not associated with the technical and ethical difficulties of installing software on the devices used by young people (*Constine, 2019*). Although this method comes with its own limitations, it introduces a more diverse picture of digital technology use than is present in the academic literature. Further, it can aide researchers disambiguating the effects of different levels of engagement and effects arising from engagement at different times of day—for example, before bedtime. In a policy landscape where parents and caregivers are increasingly advised to limit adolescent digital engagement on the basis of research done using retrospective self-report data, diversifying measurement by including time-use diary measurements is an increasingly exciting scientific prospect.

### The present research

In this study we investigated the association between digital screen engagement and adolescent sleep using a unique high-quality birth cohort dataset that includes both retrospective self-report measures of digital engagement, time-use diary data, as well as sleep duration and bedtimes data. We used these data to create additional measures of bedtimes and digital technology use both throughout the day and in the 30 min period before bedtime. Furthermore, both the retrospective self-report and diary measures were available for weekdays and weekend days separately. Briefly, our study aimed to explore three research questions: first, how does digital technology use relate to sleep onset and duration on weekdays? Second, How does digital technology use relate to sleep onset and duration on weekends? And third, How is the relationship between digital technology use and sleep affected by the measurement practices used to quantify digital technology use?

To fully explore these relations arising from our research questions we apply Specification Curve Analysis (SCA; *Simonsohn, Simmons & Nelson, 2019*). This analytical method allows us to determine the extent to which the availability, use, and time of use of digital technologies impacts teen's sleep outcomes, while taking into account the range of possible analytical decisions we could have made, and the analytical pathways we could have taken to ultimately analyze the data. Our goal is to provide a transparent overview of the effects found in the data available, supporting robust inferences regarding the nature of the relations between digital technology use and adolescent sleep.

## MATERIALS AND METHODS

### Dataset

The research data used in this study was drawn from the Millennium Cohort Study (MCS), a nationally representative longitudinal birth cohort study which tracks a cohort of young people as they live their lives in the UK (*University of London, Institute of Education, Centre for Longitudinal Studies, 2017*). The sample includes young people who were born between September 2000 and January 2001 and the sampling frame oversamples minorities and those from disadvantaged backgrounds to allow for direct comparisons between those who do and do not suffer material deprivations. We did not use weightings in our analyses. Furthermore, we did not impute missing data and instead used listwise

deletion. The MCS data we analyzed in this study originated from two aspects of the project. First, retrospective data was extracted from the omnibus self-report survey which was administered in 2015 and 2016; A total of 11,884 adolescents (5,931 girls and 5,953 boys) completed this aspect of the study. Their ages ranged from 2,864 13-year-olds, to 8,860 14-year-olds and 160 15-year-olds. Second, we used data from the time-use diaries which 4,642 adolescents, a subsample of the omnibus participants, completed. Not all participants could fill out the time use diaries due to restrictions in the amount of activity monitors available that were administered concurrently, but whose data we do not evaluate in this study (*Ipsos MORI, 2016*). In other words, the subsample of those who completed time-use diary data is about one third of the sample available for analyses of only retrospective self-report data. The participants who completed time-use diary data differ significantly from those who did not in terms of age ($m_{\text{not completed}} = 13.8$, $m_{\text{completed}} = 13.7$), gender ($m_{\text{not completed}} = 0.53$, $m_{\text{completed}} = 0.45$), closeness to parents ($m_{\text{not completed}} = 3.20$, $m_{\text{completed}} = 3.23$), retrospectively-reported screen time ($m_{\text{not completed}} = 4.85$, $m_{\text{completed}} = 4.76$) but not self-reported sleep duration on weekends ($m_{\text{not completed}} = 10.55$, $m_{\text{completed}} = 10.51$) and weekdays ($m_{\text{not completed}} = 8.61$, $m_{\text{completed}} = 8.64$). The participants who completed the time use diaries were more likely to be younger, female, closer to their parents and use less screens.

### Ethical review

Ethical approval for the MCS was given by the UK National Health Service (NHS) London, Northern, Yorkshire and South-West Research Ethics Committees (MREC/01/6/19, MREC/03/2/022, 05/MRE02/46, 07/MRE03/32). Parents gave written consent, while adolescents provided oral consent.

### Measures

In this study we considered measures of bedtime derived from time use diaries and retrospective self-report questionnaires, in addition to measures of sleep duration and sleep difficulties derived solely from retrospective self-report questionnaires. Likewise, we made full use of the available technology engagement data collected in both the retrospective and time-use components of the MCS.

The time-use diaries were developed specifically for the MCS by Ipsos MORI and the Centre for Time Research at the University of Oxford and included two 24 h time windows (weekday and weekend) where the cohort members reported on what they are doing throughout the day (*Ipsos MORI, 2016*). They were administered in the 10 days after the interviewer visit where the adolescent filled out the questionnaires and were available on the web, through an app or on paper. The web- and paper-based versions were designed in such a way that participants reported on 10 min intervals filled out throughout the day from 4 AM to 4 AM the next day. On the app, however, they could drag and drop activities up to a minute's interval for that time frame. When recording their activities, the adolescents were provided with a wide variety of 44 activity codes to choose from; one code related to sleep, while five codes related to digital engagement, we will detail these

codes in the time-use diary sections below. The time-use diaries could also be filled in retrospectively, for example, when noting sleep times.

### Retrospective self-reports of sleep

In our main analyses we examined measures of bedtime and sleep duration on weekend days and weekdays using retrospective self-report questionnaires. In additional analyses we also examined sleep difficulties. Bedtime on weekdays was measured by a self-report question which asked "About what time do you usually go to sleep on a school night?", for weekend days it asked "About what time do you usually go to sleep on the nights when you do not have school the next day?": 1 = "before 9 PM," 2 = "9–9:59 PM," 3 = "10–10:59 PM," 4 = "11-midnight," 5 = "after midnight." These scores were reversed so that higher scores mean earlier bedtimes. The mean for bedtime on weekdays was 3.06 (SD = 0.96) and 2.04 (SD = 0.93) on weekends. While these measures were on a 5-point scale, we treated them as continuous because it was important for us to apply the same analytical approach to all specifications (analytical pathways) of this research.

To calculate total time spent sleeping on weekend days or weekdays we took into account the reported wake up times. Wake up times were measured on weekdays using the following question: "About what time do you usually wake up in the morning on a school day?", 1 = "before 6 AM," 2 = "6–6:59 AM," 3 = "7–7:59 AM," 4 = "8–8:59 AM," 5 = "after 9 AM." On a weekend day, wake up time were measured using the question: "About what time do you wake up in the morning on the days when you do not have school?", 1 = "before 8 AM," 2 = "8–8:59 AM," 3 = "9–9:59 AM," 4 = "10–10:59 AM," 5 = "11–11:59 AM," 6 = "after Midday." To obtain the total time sleeping we first calculated the approximate bedtime: adding 7.5 to the adolescent's score on the retrospective self-report bedtime question. This meant, for example, that a score of 1 ("before 9 PM") became 8.5, and a score of 4 ("11-midnight") became 11.5. We therefore assumed the middle of the range selected to be the bedtime to be the actual bedtime. We calculated the amount of time spent sleeping before midnight by subtracting this bedtime value from 12. For example, for those who scored 4 ("11-midnight"), we coded their bedtime to be 11.5, and therefore their total time spent sleeping before midnight was 0.5.

We then calculated the time spent sleeping after midnight: for weekdays we added 4.5 to the score on the self-report wake up time question and for weekends we added 6.5 to the score on the self-report wake up time question. Again this assumed the middle of the range selected to be the wake up time. We then summed the time spent sleeping before and after midnight for both weekend and weekdays separately and used the measure as continuous. For the measure of total time spent sleeping on weekdays the mean was 8.62 (SD = 1.04) and on weekends it was 10.53 (SD = 1.23).

Furthermore, we also examined sleep difficulties in additional analyses. These difficulties were measured using two retrospective self-report scales concerning how long it takes for an adolescent to fall asleep and whether the adolescent wakes up during the night. In our regressions we analyzed the variables as continuous. The former is measured by asking: "During the last 4 weeks, how long did it usually take for you to fall asleep?", 1 = "0–15 min," 2 = "16–30 min," 3 = "31–45 min," 4 = "46–60 min," 5 = "more than

60 min" (Mean = 3.73, SD = 1.27). The latter is measured by asking: "During the last 4 weeks, how often did you awaken during your sleep time and have trouble falling back to sleep again?", 1 = "all of the time," 2 = "most of the time," 3 = "a good bit of the time," 4 = "some of the time," 5 = "a little of the time," and 6 = "none of the time" (Mean = 4.61, SD = 1.37).

### Time-use diary reports of sleep

We also measured adolescents' bedtime using their time-use diaries, where adolescents could indicate that they were sleeping during certain times using a specific code (they would need to fill out those areas of the time use diary retrospectively). To locate when the adolescent went to sleep in the evening we wrote a piece of code that scanned through each adolescent's time use diary and noted the last time during their specific record when their activity changed from a non-sleep activity to sleep. As the time use diary ranged from 4 AM on the study day to 4 AM on the next day this should provide us accurate bedtimes for most adolescents (i.e., those who went to bed before 4 AM). For those adolescents who did not have any sleep onset during the diary day (i.e., they went to bed after 4 AM the next day), we assumed they went to sleep at 4 AM.

We did not include any measure of sleep duration, as the estimation of this using the same day's wakeup time would have required making assumptions that we found untenable. For example, if the teenager filled the time-use diary out on a Monday, and we wanted to measure sleep duration we would only have information for their wake-up time on Monday morning and bedtime on Monday evening. We would therefore need to assume that their wake-up time on Monday morning would be identical to their wake-up time on Tuesday morning, and then use this to calculate the time between their bedtime on Monday evening and their wake-up time on Tuesday morning. As wake-up times are subject to change, we did not think the assumption that Monday and Tuesday wake-up times are always equal would be beneficial to this study.

The scale of these measures was on a continuous scale from 1 to 144 (10 min intervals throughout the 24 h day, with higher scores meaning earlier in the day). The mean for the time gone to sleep was 30.22 (~11 PM, standard deviation 14.41) on a weekday and 27.74 (~11:20 PM, standard deviation 13.81) on a weekend.

### Retrospective reports of digital engagement

The dataset included a retrospective self-report digital screen engagement item set which asked: "On a normal week during term time, how many hours do you spend… ," (a) "watching television programs or films? Please remember to include time spent watching programs or films on a computer or mobile device as well as on a TV, DVD etc. Please also include time spent before school as well as time after school" (b) "playing electronic games on a computer of games systems, such as Wii, Nintendo D-S, X-Box or PlayStation? Please remember to include time before school as well as time after school?" (c) "using the internet? Remember to include time spent using the internet on tablets, Smartphones and other mobile devices as well as computers and laptops. Please don't include time spent using the internet at school, but remember to include time before and

after school and anytime for homework," and (d) "on social networking on messaging sites or Apps on the internet such as Facebook, Twitter and WhatsApp?". For all four questions the response options ranged from 1 = "none," 2 = "less than half an hour," 3 = "half an hour to less than 1 h," 4 = "1 h to less than 2 h," 5 = "2 h to less than 3 h," 6 = "3 h to less than 5 h," 7 = "5 h to less than 7 h," to 8 = "7 h or more." To obtain a composite digital screen engagement items we took the measures' means, deleting those adolescents who had missing values in one or more of the four items. The mean of the scores on the composite measure was 4.82 while the standard deviation was 1.26. The measure was treated as continuous.

### Time-use diaries of digital engagement

Digital technology use was also measured in the time-use diaries as adolescents could use five different codes to signify time spent using technologies: "answering emails, instant messaging, texting," "browsing and updating social networking sites," "general internet browsing, programing," "playing electronic games and Apps" and "watching TV, DVDs, downloaded videos." These five codes were aggregated to form the digital engagement score, which was then split into three distinct measures of digital technology use: (1) Participation—whether the adolescent engaged in any digital screen engagement during the day, (2) Time spent—for those adolescents who did participate in digital screen engagement, we measured the total time spent on such activities, and (3) Bedtime technology use—whether digital technology was reported to be used 30 min before the bedtime reported in the time-use diaries. Like with the sleep measures, these measurements were separated for weekdays and weekend days. For the dichotomous participation measure on weekdays the mean was 0.81 and the standard deviation 0.39; on weekends the mean was 0.85 and the standard deviation 0.35. For the continuous time spent measure on weekdays the mean was 3.45 and the standard deviation 2.49, while on weekends the mean was 4.59 and the standard deviation 2.98. Lastly, for bedtime technology use on weekdays the mean was 0.46 (SD = 0.50) and on weekends the mean was 0.47 (SD = 0.50).

We bifurcated time-use diary measurements into participation and time spent because the raw values showed high positive skew. In other words, while many participants did not register engaging in any digital technology use throughout the day, very few participants registered a very large amount of digital technology use. In accordance to previous studies (*Orben & Przybylski, 2019a*; *Rohrer & Lucas, 2018*), we therefore split the time use variable into participation and time spent (*Hammer, 2012*; *Rohrer & Lucas, 2018*). For amount of technology use before bed we chose to use a dichotomous measure examining 30 min before bedtime because this would include only proximal technology use before bed. We could have also included longer before-bed timeframes (e.g., 1 or 2 h) or used continuous measures of how much technology was used in such longer timeframes. We decided against this because of the binned nature of the time-use diaries. That said, we acknowledge that such choices could have been included in a more elaborate specification curve model.

### Covariates and confounding variables

We included a variety of covariates in the analysis of this data, having chosen these on the basis of existing theory and past studies (*Orben & Przybylski, 2019b*; *Parkes et al., 2013*; *Przybylski, 2018*). The control variables spanned maternal, family, child and demographic factors. Firstly, the demographic factors included the sex and age of the child and the ethnicity of the mother. Sex was coded dichotomously with one being male and zero being female, the age of the child was a continuous variable and the ethnicity of the mother was coded as a factor, white (1) and nonwhite (0), to assess the ethnic majority/minority. We did not take child ethnicity, because we were concerned about the quality of responding as the participants were asked about ethnicity at a young age. We decided parent ethnicity would be a more consistent and high-quality measure.

Next the family-level control variables included weekly family income, whether the father lives with the family and the number of siblings in household. These are important to control for as the family environment could causally affect both sleep and screen habits. Weekly family income and number of siblings in household were continuous measures provided by the parent while the adolescents were asked whether their father lives with them, which was coded as a dichotomous variable (No = 0, Yes = 1). The mother-level control variables included parent word activity score, highest academic qualification, time spent with child and the caregiver's score on the Kessler depression scale. These control variables were included to account for the home environment and parenting, which could both influence screen use and sleep. The word activity score is a measure of cognitive capacity and provides a continuous score from 0 to 20. The highest academic qualification was measured from National Vocational Qualification Level 1 to 5, we coded foreign qualifications or those who answered "none of these" as NA: it was coded as a factor. Time spent with child was measured on a 5-level continuous scale: "too much time," "more than enough time," "just enough time," "not quite enough time," "nowhere near enough time." The final family-level factor, parent scores on the Kessler depression scale were included, ranged from 0 to 24, and was treated as a continuous control variable.

Lastly, additional child-level control variables included closeness to parents, long-term illness and educational motivation. These items are included to attempt to control for both parenting and other factors that could influence either of the variables of interest. We included long-term illness, and no other well-being scores, because we wanted to account for debilitating aspects in the child's life but felt that general well-being items could be mediators of the relationship between screen use and sleep. Closeness to parents was measured with four four-level questions asking how close the child was with both mother and father and also how often they argued with mother and father respectively. If the child reported not having a father or mother, we coded them as NA. Educational motivation was measured using a six item questionnaire asking "How often do you try your best at school," "How often do you find school interesting," "How often do you feel unhappy at school," "How often do you get tired at school," "How often do you feel school is a waste of time," "How often difficult to keep mind on work at school." The two first

questions were reversed and the mean of the six items were taken: those with missing values for any of the questions were marked NA.

## Analytic approach

Because the topic of our study is of keen interest in both basic and applied scientific arenas, we implemented an innovative statistical approach to ensure that high standards of analytical transparency and rigor were followed to provide reliable answers to our research questions. We would have preferred to preregister our analysis protocol, a simple way to improve quality of analysis and reporting (*Wagenmakers et al., 2012*) which has been applied to research using retrospective reports to investigate sleep outcomes (*Przybylski, 2018*). We, however, had accessed the data prior to our formulating our research questions, making it impossible to document our lack of foreknowledge about the data under analysis (*Weston et al., 2019*). The analyses were therefore centered around an SCA approach (*Simonsohn, Simmons & Nelson, 2019*), which has been applied recently in work both outside the area of digital media effects research (*Rohrer, Egloff & Schmukle, 2017*) and in digital media effects research specifically (*Orben, Dienlin & Przybylski, 2019*; *Orben & Przybylski, 2019a*, *2019b*). Instead of reporting only one result of a single possible analysis pathway, as done in most psychological studies, SCA takes into account the vast number of analytical decisions associated with the analysis of data (*Steegen et al., 2016*). These decisions quickly branch out into a vast "garden of forking paths": a combinatorial explosion in the ways that data could have been analyzed. The choice between many different analysis pathways can skew the final results reported in a final article (*Gelman & Loken, 2014*; *Orben & Przybylski, 2019b*). Using SCA, we ran and reported the results of all theoretically defensible analysis pathways, in a stepwise methodological approach which is elaborated on further below.

## Step 1. Preliminary analyses

The first analytical step of our study was not related to SCA but, instead, was informative in providing a fuller picture of the available data. We first focused on examining retrospectively self-reported digital technology use, in particular by examining the correlations between retrospectively-reported digital engagement and retrospectively-reported sleep items. Examining the retrospective self-report questions allowed us to obtain an overview of what most studies in the research area are confronted with when analyzing their data. This allowed us to put our results into the context of the previous literature (*Scott, Biello & Woods, 2019*).

We also examined the correlation between self-reported retrospective digital technology engagement and sleep difficulties. Sleep difficulties were measured using two questions that asked the adolescent participants for retrospective self-report judgements about the time it takes them to fall asleep and the amount of times they awake during the night. To determine how retrospective reports of digital technology use affects these judgements, we fitted a linear regression predicting sleep difficulties from digital technology engagement and included all the covariates also used in the SCA detailed in the next methodological subsections.

## Step 2. Identifying specifications

The next step in the analytical process was to decide which specifications—combinations of analytical decisions or paths through the analytical "garden of forking paths"—are theoretically defensible and should therefore be included in the SCA. First, we noted what analytical decisions needed to be taken to analyze the data, and then, for each decision, agreed on possible analytical choices that would be plausible approaches if applied to the data by other researchers interested in our research questions. For example, an analytical decision might be how to define digital engagement and the choices might include a range of possible retrospective self-report technology use items and time-use diary variables pertaining to whether technology was used and when. For the current research question, the analytical decisions included (a) Whether to examine weekdays or weekend days? (b) How to measure digital engagement? (c) How to measure sleep? and (d) What control variables to include on the basis of the existing literature? Once the possible analytical options were determined, we proceeded to step 3 of the analysis plan and implemented all possible analytical pathways in line with each possible combination of the chosen operationalizations we identified.

## Step 3. Implementing specifications

We therefore combined the various analytical decisions, and their corresponding choices, to form the specifications of our SCA. In order to account for the variety of different analytical pathways that could have been used to analyze the data, we took all the possible combinations of analytical choices and added them into a linear regression as different standardized predictors, outcomes and covariates. In other words we analyzed every possible variant of the outcome variable in combination with every possible version of the predictor and every possible version of the covariates. Once we calculated all the resulting standardized regression coefficients, we examined their range of effects, while also taking note of their corresponding $p$ values, number of participants and $r$-squared values. We also calculated 500 bootstrapped SCAs to obtain 95% Confidence Intervals. All our calculations were then visualized in a graph presenting the range of possible standardized regression coefficients from all specifications and what analytical decisions caused which specified outcome. Said differently, the visualization helps us understand what analytical decisions were most influential in determining the nature of the analytical result and what median association linked digital technology engagement and adolescent sleep.

It is not possible to apply standard parametric significance tests to the results of the SCA analysis due to the non-independent nature of the separate specifications. In a move away from significance testing to a more in depth interpretation of the data, we decided against calculating overarching non-parametric significance estimates as implemented by *Orben & Przybylski (2019a*, *2019b)*.

## Code availability statement

The code used to analyze the data in the manuscript is available on the Open Science Framework: https://osf.io/jhkt6/.
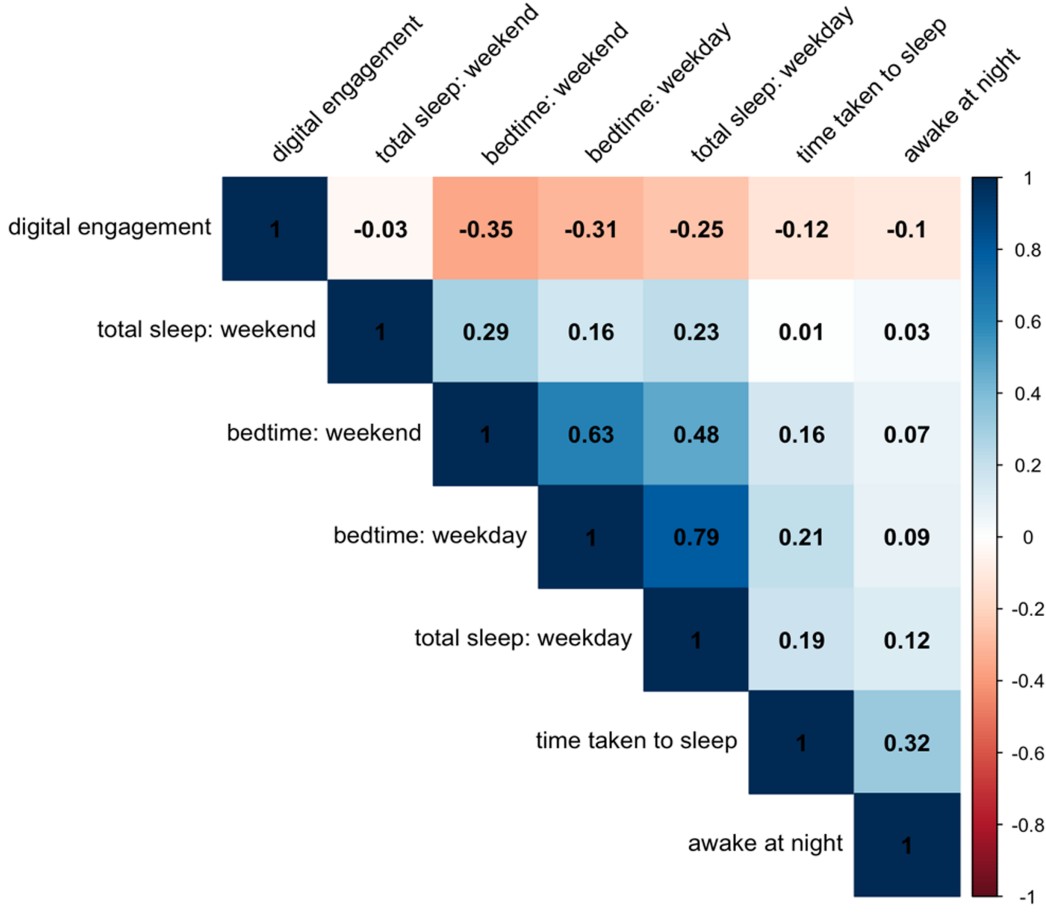

**Figure 1 Correlation matrix showing the correlation between various self-report sleep measures and self-reported digital engagement.** Red indicates negative correlations while blue indicates positive correlations.

## RESULTS

### Step 1. Preliminary analyses

We first examined the Pearson correlations between retrospective self-reported technology and sleep measures (see Fig. 1). These relations are especially informative because such measures are normally the exclusive focus of research into digital engagement and sleep. We found that there were a wide range of correlations between self-reported digital technology engagement and sleep measures ($r = -0.03$ to $-0.35$, all Confidence Intervals do not include zero). We also observed correlations between self-reported retrospective digital technology engagement and self-reported sleep difficulties: higher intensity technology users reported taking longer to fall asleep ($r = -0.12$) and that they awake more often night ($r = -0.10$).

Because these two variables are not specific to either a weekday or weekend, we could not include them in our main analyses; instead we performed two linear regressions to test their significance when accounting for all the control factors included in the main SCAs. For time taken for the adolescent to fall asleep, there was a significant negative

**Table 1 Table showing the four different analytical decisions that needed to be taken to analyze the data.** The decisions were: what day of the week to analyze, how to define sleep, how to define technology use, and which control variables to include. Each analytical decision had multiple analytical options that could have been taken. Each unique combination of different analytical options that could be implemented to analyze the data is a specification. In total, this study encompassed 120 defined specifications.

| Analytical decision | Day of the week | Sleep | Technology use | Control variables |
|---|---|---|---|---|
| Analytical options | Weekday | Bedtime (retrospective) | Participation (time-use diary) | Demographics |
| | Weekend day | Bedtime (time-use diary) | Before bedtime (time-use diary) | Demographics + child-level |
| | | Total time sleeping (retrospective) | Total time spent (time-use diary) | Demographics + mother-level |
| | | | Total time spent (retrospective) | Demographics + family-level |
| | | | | All control variables |

association: standardized $\beta = -0.067$, SE = 0.011, $p < 0.001$. However, for sleep disruptions during the night, there was no significant association, speaking against the generalized fears that digital technology leads to more night waking: standardized $\beta = -0.015$, SE = 0.011, $p = 0.167$.

## Step 2. Identifying specifications

The second part of the analysis was to determine the analytical decisions necessary to analyse our data to answer the research questions. Each analytical decision could be addressed using multiple analytical options. Determining these options in turn determined the specifications we implemented in the SCA (see Table 1 for overview). We examined relations on a weekend day and weekday separately. In our analyses we could focus on three different types of sleep measures: (1) When the adolescent went to bed—higher scores indicative of an earlier bedtime—measured using retrospective self-reports, (2) When the adolescent went to bed measured using time-use diaries, and (3) Total time spent sleeping measured using retrospective self-reports. To measure technology use, we could use three possible measures taken from time-use diaries: (1) Participation, whether the adolescent mentioned any digital technology engagement in their daily diary, (2) Time spent, when they did mention digital technology engagement, we calculated how much time they spent on digital devices, and (3) Bedtime technology use, whether they noted down digital technology engagement 30 min before bedtime. We could also use a self-report retrospective measure of digital technology use. While this measure asked about adolescent's digital engagement on a normal school day, we also used it as part of the weekend SCAs because its importance as a variable eclipsed this issue of scope.

The other analytical decision which needed to be accounted for was the inclusion of covariates. While previous studies (*Orben & Przybylski, 2019b*, *2019a*) included a "no control variables" option in their SCA models, the current study did not include this option because we did not believe the no control option is "theoretically defensible" as an analytical specification in light of simple control variables being routinely included in epidemiological and psychological research (*Simonsohn, Simmons & Nelson, 2019*) and those specifically found to be critical for interpreting media effects research on sleep (*Przybylski, 2018*). We however included either only demographics, demographics and

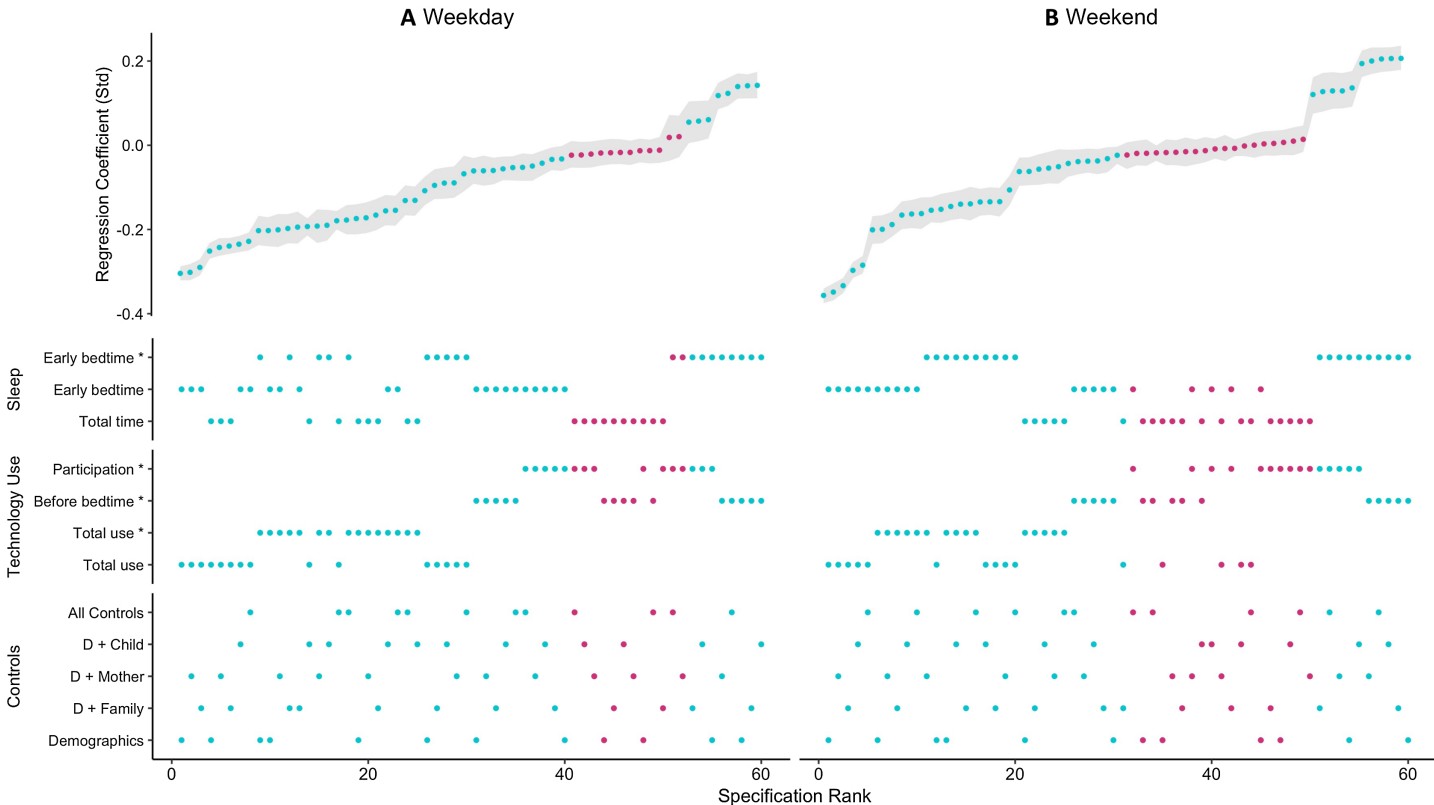

**Figure 2 Specification curve analysis.** SCA showing the results of the two separate SCAs for weekdays (A) 60 specifications and weekend days (B) 60 specifications with standardized regression coefficients presented in ranked order ranging from those results with the most negative regression associations to those with the most positive ones. In the top half of the graph the resulting standardized regression coefficient is shown. In the bottom half of the graph one can read off the analytical decisions that constitute the specification that results in the corresponding standardized regression coefficient (* = analytical variables calculated using time-use diaries). Teal dots represent statistically significant specifications ($p$-value < 0.05) while pink dots represent non-significant specifications ($p$-value > 0.05).

child-level control variables, demographics and mother-level control variables, demographics and family-level control variables or all control variables.

## Step 3. Implementing specifications

We identified 120 promising combinations of analytical decisions that we could use to test our research questions. The results of implementing these SCAs are visualized in Fig. 2, with weekend (A) and weekday (B) of the SCA separated for comparison. The number of participants for each specification is displayed in Fig. 3. The standardized regression coefficients that resulted from the various specifications ranged from the most positive (standardized $\beta = 0.21$, specification: weekend day/using technology before bedtime/ bedtime measured using time use diaries/demographic controls only) to the most negative (standardized $\beta = -0.36$, specification: weekend day/self-reported technology use/self-reported bedtime/demographic controls only). About one quarter of the specifications ($k = 31$) were not statistically significant, with the weekday SCA having 12 non-significant results and the weekend SCA having 19 non-significant results. Furthermore, there were eight specifications for weekdays and 10 specification for weekend days that were

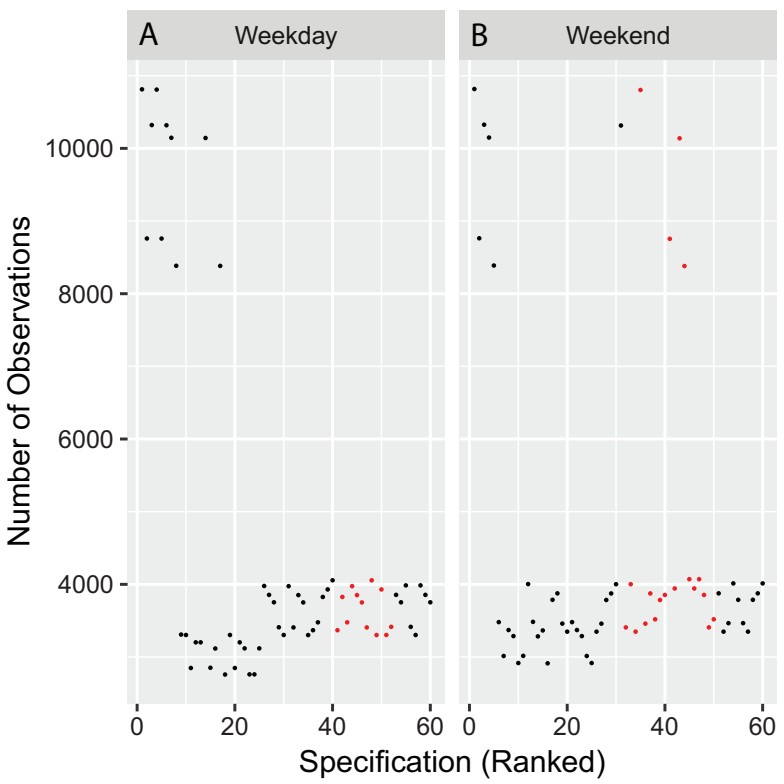

**Figure 3 Number of participants included in each specification.** Each point shows the amount of participants included in the analysis of each specification for both weekdays (A) and weekends (B) visualized in Fig. 2.               

significant and positive. The majority of specifications, however, demonstrated negative associations between digital technology use and adolescent sleep.

To examine our two research questions probing the relation between digital technology use and sleep measures, we differentiated further between weekday and weekend days. Overall weekday digital engagement had a more negative association with sleep measures than weekend engagement (median standardized $\beta_{\text{weekday}}$ = −0.06, median standardized $\beta_{\text{weekend}}$ = −0.03). This difference is significant, as shown when applying a $t$-test comparing weekday and weekend day specifications ($t_{60}$ = −5.00, $p$ < 0.001).

Looking at different sleep measures specifically, there were no apparent differences between relations on weekdays or weekend days when bedtime was measured using time-use diary or retrospective self-report measures. Bedtime was delayed when adolescents used more digital technology on both weekdays (RQ1, median standardized $\beta_{\text{time-use diary}}$ = −0.02, median standardized $\beta_{\text{retrospective self-report}}$ = −0.11) and weekend days (RQ2, median standardized $\beta_{\text{time-use diary}}$ = 0.01, median standardized $\beta_{\text{retrospective self-report}}$ = −0.10). Interestingly, however, while bedtimes on weekdays and weekend days were equally associated with digital technology use, total time sleeping demonstrated a more negative association with digital technology use on a weekday when compared to a weekend day (median standardized $\beta_{\text{weekday}}$ = −0.08, median standardized $\beta_{\text{weekend}}$ = −0.02).

This finding might support the hypothesis that negative technology effects are exacerbated on weekdays because on these days the wake-up time is static for adolescents because of school, and therefore a belated bedtime makes them lose more sleep (H. Scott, 2019, personal communication).

To answer our third research question, we examined different types of digital technology use measures. Most prominently, total time spent using digital technology reported via retrospective self-report questionnaires (median standardized $\beta_{weekday} = -0.23$, median standardized $\beta_{weekend} = -0.13$) and time use diaries (median standardized $\beta_{weekday} = -0.18$, median standardized $\beta_{weekend} = -0.15$) showed the most negative associations with adolescent sleep. In contrast, participation in digital technology use (median standardized $\beta_{weekday} = -0.02$, median standardized $\beta_{weekend} = 0.01$) and use before bedtime (median standardized $\beta_{weekday} = -0.02$, median standardized $\beta_{weekend} = -0.02$) showed a mix of very small negative and positive correlations, or non-significant results. The way technology use is measured is therefore important to consider in the study of these associations. In contrast, we also examined how the use of different control variables influenced the results, but there were no clear trends of some control variables shifting results in a specific direction (median standardized $\beta = -0.05$ for all specifications of control variables).

## DISCUSSION

Concerns regarding adolescents' digital engagement and its possible influence on sleep are at the forefront of popular and policy discourse (*Davies et al., 2019*; *Viner, Davie & Firth, 2019*). Unfortunately, the existing evidence base relies largely on imprecise measurements of digital engagement (*Orben, Etchells & Przybylski, 2018*; *Orben & Przybylski, 2019a*). A more nuanced understanding of digital engagement and its relationship to adolescent sleep grounded in transparent methods using a convergent and accurate measurement approach is necessary. This need is pronounced as most previous research does not dissociate digital engagement throughout the day from digital engagement before bedtime, but most policies and draft guidance on this topic specifically make reference to digital engagement before bedtime (*Davies et al., 2019*; *Viner, Davie & Firth, 2019*). Guidance provided to caregivers and policymakers therefore fails to align with the evidential value of the research available. In light of this gap, our study introduces multiple improvements to this important research area by diversifying measurement and improving analytical transparency whilst probing digital engagement before bed. In this study we included both retrospective self-report measures and time-use diary measures to examine the association between digital engagement and adolescent sleep using SCA. The time-use diaries allow us to dissociate different measures of digital engagement: providing some of the first insights based on time-use diaries into the use of digital technologies before bedtime and whether this affects adolescent sleep outcomes.

In terms of our first two research questions, we investigated the association between digital engagement and sleep on both weekdays and weekend days. We found evidence for negative associations between digital engagement and adolescent sleep on both weekdays ($\beta = -0.06$) and weekend days ($\beta = -0.03$) and found evidence that the relation on

weekdays, though itself modest, is significantly more negative than the relation on weekend days. It is interesting to note, however, that there exist divergent results when examining different measures of adolescent sleep. When we examined total time spent sleeping, we found it was more negatively associated with digital technology engagement on a weekday rather than a weekend day. Such a difference was not present when we examined bedtimes: the negative association between bedtime and digital engagement was similar for both weekdays and weekend days.

This pattern could be, in part, due to wake-up times being less flexible on a weekday, where adolescents need to wake up at a certain time for school regardless how late they stayed up at night. Because adolescents have less choice regarding their wake-up time on weekdays, they might be accruing a sleep deficit when their bedtime is forestalled by technology use. On weekend days they are often able to choose to wake up later, therefore losing less total sleep. This highlights the importance of differentiating weekdays and weekend days when investigating adolescents' technology engagement and its relationship to sleep.

It is, however, crucial to note that the effects on weekdays and weekend days were both very small, with an effect size that falls into the range of practically insignificant effects previously defined by media scholars (*Ferguson, 2009*; *Orben & Przybylski, 2019a*; *Przybylski, Orben & Weinstein, in press*). Whilst the standardized estimates we observe will be of interest to basic science researchers (e.g., to plan their sample sizes), these associations are possibly too small to be of great importance for policy makers or parents. While it is difficult to put the effects into context due to the many different response scales, we will do so by focusing on the hourly outcome measures we analyzed, namely the total time spent sleeping which was derived from retrospective self-report scales and the technology use before bedtime and total technology use throughout the day (derived just from time-use diaries) and all the control variables included. Doing so indicated that every additional hour of technology use throughout the day was associated with 9 min and 17 s less sleep on weekdays and 3 min and 26 s less sleep on weekend days. This result conceptually replicates findings reported from a pre-registered representative cohort study of American children and adolescents which estimated the impact of an hour of screen time on sleep ranges 3–8 min of sleep depending on the age of the child (*Przybylski, 2018*).

The modest scale of this association is similarly reflected in digital screen time before bed. Results here showed that an adolescent who reported using digital technology within a half an hour before bedtime reported an average of 1 min less sleep on weekdays and 1 min and 3 s less sleep on weekend days. Framing these small effects in terms of the real-world units parents and health policy-makers care about is important; they speak to the idea that technology use before bed might not, in of itself, be associated with the practically significant negative impacts often in much media coverage and policy debate.

Keeping this in mind, it is important to make note of findings relating to our third research question. Most negative associations are generally found when examining retrospective self-report measures of digital engagement. As these are the measures most often examined in previous research, this is further evidence that the heavy reliance on retrospective self-report measures presents a pronounced problem for the research area.

Results suggested this is problematic especially because measures of technology use before bedtime were not meaningfully associated with sleep outcomes, even though bedtime technology use is often hypothesized to be an important driver of lack of sleep. The exclusive use of retrospective total screen time measures might, therefore, be skewing the literature towards over-estimates of associations than would be found if more diverse, potentially more accurate, measurement methodologies were implemented.

Considering the implications of the present study more broadly, the pattern of results we report in this investigation makes it is increasingly evident that screen time as a concept is flawed, potentially fatally so. Retrospective reports of screen time fall well short of accounting for the sheer diversity of behaviors the concept and measures it stands in for. It is clear to many that 20 min on very social digital content, possibly co-using the technology with parents or friends, will have a very different influence on a child than 20 min on digital content that can contain harmful or disturbing information or images. It is therefore important for future research to incorporate a greater variety of measurement into its designs, moving away from relying solely on retrospective self-report accounts of screen time.

## Limitations

While this study excels at introducing new measurement to the area of technology effects and sleep research, it is important to note three limitations that need to be taken into account before generalizing results. Firstly, while the use of time-use diaries allows us to probe more diverse measures of digital engagement, it is not known if the specific measures we are using in these data necessarily contain less error than the common retrospective self-report measures. It might be the case that adolescents are simply not accurate in reporting digital engagement by way of time-use diaries either and how these instruments are misused is merely different. The errors in the two different measurement approaches will most probably vary, making it valuable to examine them in conjunction, as done in this study. That said, research specifically focused on how digital engagement is recorded in time-use diaries to gauge what sort of error is associated with this specific technique is needed to advance this topic of study.

Second, while our study innovated in terms of time-use diaries, the secondary nature of the data meant that some of the self-reported measurements had limitations attached to them. A more continuous measure would have also benefited our report of adolescent's self-reported bedtimes, which was measured on a 5-point scale but which we took to be a continuous variable for the scale of analytical clarity. The nature of the time-use diaries also made it necessary to quantify things in ways that introduced other limitations. Digital technology use before bed was operationalized as noting down use of digital technologies in the 30 min before the noted bedtime in the time use diaries. Again, a more continuous measure once new studies are finalized could benefit understanding further. When examining sleep using the time use diaries, we also needed to assume that the last sleep onset during the diary day was "bedtime," which could not be the case in adolescents with disturbed sleep. Furthermore, for those adolescents who never noted a sleep onset during the diary day, we had to implement the assumption that they went to bed at 4 AM.

We therefore report further sensitivity analyses in the supplementary code. Lastly the nature of the data was limited to measuring time spent on screens and as noted above screen time is known to be a flawed measure of digital technology effects and should be replaced in future research by measures that take into account the diversity of technology use.

Finally, in an area where there is much public debate and academic discussion, it is also important to highlight the correlational nature of this study. The results should therefore not be used to make directional or causal conjectures. As there are currently no multi-wave time-use diary studies that would have allowed us to examine diverse measures of digital engagement, this limitation is a clear tradeoff as we cannot examine longitudinal within-person effects. In future, more longitudinal data collection with diverse measurement methods would allow for directional conjectures to be made.

## CONCLUSIONS

The widespread use of digital technologies by young people and its implication on sleep has been widely discussed in both academic, care giving and political circles. If the use of digital screens either throughout the day or before bedtime undermines sleep quality, duration, or causes a shift in bedtimes, this needs to be noted and addressed quickly. In this study we found that digital technology use is negatively correlated with sleep measured on both weekends and weekdays, however these correlations were small. While there were not many pronounced differences between the relations on a weekday compared to a weekend day, weekdays showed a more negative association between digital technology use and total time spent sleeping, something that could hint at the rigidity of waking times on school days. It is interesting to note that no clear associations were found when examining digital technology use before bedtime specifically, and that the negative associations were driven by retrospective accounts of digital technology use or total time spent using digital technologies as reported in time use diaries. This highlights the importance of thinking about how digital technology use is measured, even though in the current scientific landscape the scope and nature of measurement are often overlooked (*Flake & Fried, 2019*). Diversifying the measurement of digital engagement is crucial to test the robustness of the effects found in previous research. Our study using time-use diaries in conjunction with transparent analysis methods marks a valuable first step towards enabling such measurement diversification, allowing researchers to examine different measures of digital engagement and sleep that have long been neglected in a new light. In future, policy makers, academics and industry need to collaborate in a more concentrated effort to further improve the measurement of digital engagement, for example, by finding ways to share individual trace data currently housed on servers of social media and technology companies. Only when research can fully remove itself from the problematic self-report measurement of digital technology use, will it be able to provide clear insights into how digital engagement is affecting current adolescents—and, ultimately, the whole global population.

## ACKNOWLEDGEMENTS

Centre for Longitudinal Studies, UCL Institute of Education collected MCS and the UK Data Archive/UK Data Service provided the data; they bear no responsibility for its analysis or interpretation.

### Funding

The Millennium Cohort Study is funded by grants from Economic and Social Research Council. Amy Orben and Andrew Przybylski were funded by a grant from Barnardo's UK, Amy Orben received funding from Emmanuel College, University of Cambridge, and Andrew Przybylski received funding from the Huo Family Foundation. The funders had no role in study design, data collection and analysis, decision to publish, or preparation of the manuscript.

### Grant Disclosures

The following grant information was disclosed by the authors:
Economic and Social Research Council.
Barnardo's UK.
Emmanuel College, University of Cambridge.
Huo Family Foundation.

### Competing Interests

The authors declare that they have no competing interests.

### Author Contributions

- Amy Orben conceived and designed the experiments, analyzed the data, prepared figures and/or tables, authored or reviewed drafts of the paper, and approved the final draft.
- Andrew K. Przybylski conceived and designed the experiments, authored or reviewed drafts of the paper, and approved the final draft.

### Human Ethics

The following information was supplied relating to ethical approvals (i.e., approving body and any reference numbers):

Ethical approval was granted by the U.K. National Health Service (NHS) London, Northern, Yorkshire and South-West Research Ethics Committees (MREC/01/6/19, MREC/03/2/022, 05/MRE02/46 and 07/MRE03/32).

### Data Availability

The data we used is available on the UK data service: https://beta.ukdataservice.ac.uk/datacatalogue/series/series?id=2000031. To access the data, users have to register with the UK data service and agree to an End User License which is an outline of the terms and conditions for using the data. Some of the data used in our study might not be accessible to

users outside the UK. Please check the terms and conditions for each of the datasets prior to use on the UK Data Service.

The code used to analyze the data is available on the Open Science Framework: Orben, Amy, and Andrew K. Przybylski. 2019. "Teenage Sleep and Digital Technology Engagement Across the Week." OSF. December 27. osf.io/jhkt6.

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
