# Peer review of "Teenage sleep and technology engagement across the week"

_PeerJ, doi:10.7717/peerj.8427_

## Round 0.1 · original submission · Major Revisions

I have received reviews from Michaeline Jensen and one other expert in the field. Both reviewers were quite positive about the manuscript and see a clear path to publication. However, both highlight areas of the manuscript that are in need to touch-up to enhance their clarity and ask for additional detail in some places. Both reviewers would like you to include more detail on your measures, specifically, and a more robust acknowledgment of the limitations of this research.

I look forward to reading a revision of this work.

·

Basic reporting

1. Overall, the authors do an excellent job of conveying complicated methods clearly and concisely, and interpreting their results in a manner that is appropriate to the method.
2. Measurement -
a. In general, I found myself looking for more details in the measurement section, to better understand the nature of the many measures used and the extent to which they were endorsed by the participants. I would have appreciated perhaps a table with means/standard deviations or even just the average rates for the different types of measures included in the text body itself.
b. Line 232: The authors state that, they did not include measures of sleep duration as this would require assumptions they found untenable. Given the overall focus of the paper on including all possible specifications, I would like more detail on why this was untenable.
c. It is admirable that this study includes such a wide variety of individual and family covariates and potential confounding variables. However, I would have appreciated a bit more information/justification on why these variables specifically were included, and more detail on what these variables measure and how. This is especially true for: parent word activity score (what does this measure? Why might it be a confound?), mother ethnicity (why not child ethnicity? How is this coded?), child long term illness (why is this included but other variables relevant to child wellbeing are not?).
3. A few typos and wording issues were noted.
a. Lower case standardized bweekend rather than β in the abstract.
b. Introduction line 61: “This engagement 61 oftentimes fail to cease” (should read “fails”).
c. I would recommend rewording this sentence for clarity (line 300): “Such decisions researchers have to make to obtain an answer to their research question, quickly branch out into a vast ‘garden of forking paths’ that can skew the final results reported in a final paper”
4. Figure 1 is a straightforward portrayal of many results. I would recommend indicating in more detail which variables were gleaned from self-report vs daily diaries (perhaps using asterisks as in the other figure) to aid the reader in deciphering patterns.

Experimental design

1. I greatly appreciate the authors' commitment to open science and transparency: They've made available via the OSF all study data and materials. This will greatly aid others in potential replication or extension of this work.
2. As the authors note, the use of daily time use diaries is a significant contribution of this study over most of the literature which uses retrospective self-reports. However, in deciding to take the average levels of reports across all study days (albeit separated by weekend and weekday), there is something of a missed opportunity in that we cannot draw conclusions about same day digital engagement and that same night’s sleep. This decision not to use within-person multilevel modeling (and instead take a between-persons specification curve approach) may be reasonable given the study aims, but the implications of this decision ought to at least be mentioned and discussed in terms of future directions for research.

Validity of the findings

1. I found that the study hypotheses were tested rigorously and transparently, and that results and their interpretations were appropriate in light of the method.

Additional comments

This paper was a pleasure to read- I think SCA is a good fit for this study question, and I would imagine this work will move the field forward in thinking carefully about the ways in which technology may (or may not be) linked with sleep difficulties.

Reviewer 2 ·

Basic reporting

Overall well-written, with a clearly communicated rationale for the current study and recommendations for next steps.

Raw data are available via the UK Data Service and the analysis code is available (OSF) and clearly annotated.

Cites relevant literature throughout. The introduction summarises existing work on specifically bedtime media use (lines 90-94), but the discussion then argues that this is a gap in available literature with the current study " providing some of the first insights into the use of digital technologies before bedtime" (lines 522-524; similar at lines 619-620). This could be reworded to frame the current added contribution (of time use data and SCA) within the context of previous work (which has explored bedtime use).

Experimental design

This study adds a valuable new measurement for digital engagement via time use diaries. However, measurement of sleep parameters remains problematic and the limitations of this should be made explicit. In particular, the annotated code indicates that sleep duration has been calculated by assigning precise bedtimes and rise times based on ordinal response categories (e.g. assigning a value of 12.30am to all answers of 'after midnight' and 10.30pm to all answers of '10-10.59pm'). The available survey response categories do not allow accurate calculation of sleep durations. It should be made clear in the manuscript exactly how this measure was derived from the available responses, especially highlighting the limitations of using this rough proxy for sleep duration. Similarly, the manuscript should highlight the limitations of using the last sleep onset of the day as 'bedtime' (which may not be appropriate for those with more fragmented sleep), and also assigning midnight to any sleep onset later than this.

More generally, additional details on measures would be useful: both sleep and digital engagement self-report measures would benefit from further clarity on how they were derived (e.g. the composite score for digital engagement), how they were treated (e.g. as ordinal or continuous, and whether they met fair assumptions for this).

Similarly, the authors should mention their approach to handling missing data (e.g. listwise deletion?) and whether survey weights were used in analysis. The authors have not claimed that these analyses remain representative of the UK adolescent population, but these points can still be noted for transparency.

I would be interested to know whether the subsample (one third) of participants who gave time use data differed from the whole sample, e.g. demographic profile, self-reported sleep/media use.

The authors have provided considerable details on their use of SCA. However this is still likely to be an unfamiliar method for most readers and would benefit from even clearer communication. Including 'worked examples' could be helpful to illustrate concepts to readers (e.g. null dataset creation, bootstrapping p-value calculations). Furthermore, a large section explaining the bootstrapping and significance testing (lines 371-390) does not include citations - was the current approach based on similar previous work, or theory or is it a novel approach?

I appreciate the authors' transparency in outlining a detailed account of their approach, sharing clearly annotated code and noting the unsuitability of pre-registration for this study. I think this transparency could be further enhanced by clarifying the process of choosing possible specifications. For example, bedtime digital engagement could equally have been specified as 1 hour or 2 hours before bed, and/or as the amount of time using media in these periods rather than participation (yes/no). The same applies to other possible covariates that could have been included (or the choice of covariate grouping). Addressing this point may simply require explicitly highlighting within the manuscript that whilst these 120 specifications offer added insight (versus just one specification), they still do not provide comprehensive coverage, and that many other plausible specifications exist.

Is linear regression appropriate for predicting the ordinal outcomes? Were there any sensitivity analyses or citations that could support the suitability of this approach?

Were the correlations Spearman or Pearson (line 399 and Figure 1)? Were all reported correlation coefficients statistically significant?

Validity of the findings

The authors rightly point out the limitations of the 'screen time' construct (e.g. lines 100-102). This could be emphasised more consistently into the discussion, noting that the current analyses use an aggregated measure of different media use. This aggregates potentially very stimulating, socially driven online interactions together with potentially very passive media consumption. This limitation could be highlighted in the discussion, with the potential for future work to explore the role of content and experience of digital engagement, rather than hours per day.

Similarly, it is important to highlight in the discussion that bedtime digital engagement has been operationalised as the presence/absence of engagement in the 30 minutes before bed, rather than the time spent on this before bed. Results may differ between bedtime media users who spend more/less time on this activity.

Comparing weekdays and weekends is a valuable contribution, given the strong influence of school timetables on adolescent sleep schedules, as discussed (e.g. lines 538-544).

Additional comments

Thank you for this well-written manuscript that presents a valuable new approach to this topic and clearly highlights the issue of measurement as this field moves forward! I have provided details of areas where I feel the manuscript could be further improved, largely around providing additional clarity on the methods used and what these can and cannot yet tell us.

---

## Round 0.2 · Minor Revisions

Michaeline Jensen graciously agreed to review your revised document. You will find her comments below. There are still a few points in the manuscript that require further clarification. I suspect these changes will be quite easy to implement. I had many of the same questions voiced by Dr. Jensen. On my rereading of the piece, I noted a couple of typos: On line 529 you say "loose" when you mean "lose." A similar error shows up on line 581 where you use "loosing" when you mean "losing."

I anticipate quick acceptance of this piece after Dr. Jensen's comments have been addressed. To editorialize for a moment, I found this manuscript a pleasure to read, and I suspect it will garner some serious attention when published. It provides solid counter-evidence to prevailing wisdom about screen time and it uses an innovative technique to do so. I appreciated that your piece could also act as a primer on the logic of SCA, as the rationale behind (nearly) every decision was clearly laid out in your narrative. Brava!

·

Basic reporting

Thank you for this opportunity to review this revised manuscript. Overall, I found that the authors were quite responsive to reviewers’ suggestions, and the manuscript is improved as a result. In particular, I appreciate the added detail in the measures section, which greatly enhances the clarity of the methods. Nonetheless, and I hope the authors will forgive me here, I still have a number of questions about how some variables were extracted/calculated, and worry that readers may likewise have a hard time following elements of the procedures and how the different measures were computed (and thus how to interpret the findings):
1. I appreciate the clarification in the rebuttal letter on my point 8 (about a missed opportunity to look at within-person effects). I see that I mistakenly assumed that there were multiple time use diaries filled out on weekdays and weekends (which would have allowed for examination of within-person effects). However, upon further review I realize that nowhere in the procedure is this important detail mentioned -that adolescents filled out only two-time use diaries (one on a weekday and one on a weekend). I suspect that this missing detail contributed to my own misunderstanding and should be added to the procedures to avoid confusion on the reader’s part. In fact, the procedure for the time use diaries in general could use a bit more detail: How/when did adolescents fill out the time use diaries? Did they always fill them out at a set time of day (e.g. 9pm), or rather at bedtime? This also raises another question- given that the authors chose to use listwise deletion, I would be curious to know what percentage of youth who participated in the time-use diary sub-study contributed both weekday and weekend observations? What was the final sample size before/after listwise deletion?
2. The calculation of total time spent sleeping (from the retrospective self-report) remains unclear. The revised manuscript details “To obtain the total time sleeping we first calculated the time spent sleeping before midnight: adding 7.5 to the adolescent’s score on the retrospective self-report bedtime question. We then calculated the time spent sleeping after midnight: for weekdays we added 4.5 to the score on the self-report wake up time question and for weekends we added 6.5 to the score on the self-report wake up time question. We then summed the time spent sleeping before and after midnight for both weekend and weekdays separately and used the measure as continuous. For the measure of total time spent sleeping on weekdays the mean was 8.62 (sd = 1.04) and on weekends it was 10.53 (sd = 1.23).” Given that self-reported bedtime was reported using a likert scale (1-5), and not exact clock times, the procedure used for calculating the time spent sleeping before/after midnight is not straightforward. The code (and Reviewer 2’s comments) suggest that discrete clock times were assigned (e.g. 10:30pm for answers of 10-10:59pm)- this detail ought to be included in the manuscript text itself rather than relegate to the code (which many readers will not reference). Furthermore, the decision to add 7.5, 4.5, and 6.5 needs more explanation.
3. Time-use diary reports of sleep- the manuscript states that “We also measured adolescents’ bedtime using a measurement extracted from their time-use diaries, where one code signified the activity of sleep”. What was the nature of the “activity of sleep”? The wording now sounds as if an adolescent reports on what time they fall asleep on Monday before they fall asleep- I am having a hard time wrapping my head around how an adolescent reports on what time they fell asleep on Monday (but does not do the reporting once they wake up on Tuesday- how does one report what time they fall asleep before doing so? Or maybe they do fill out the time use diary on Tuesday, about the entire preceding day?). The inclusion of the variable wording might make this clearer, as would more details about the diary procedure (see point 1 above). Also, the revised manuscript notes that the authors “imputed the time of last sleep onset during each diary day”. The use of imputation here needs more detail. What was the procedure for imputation?
4. Retrospective self-report of digital engagement: the measures currently include the item wording as, “On a normal week during term time how many hours do you spend…” Is this question meant to report on digital engagement over the course of the whole week? Or rather on a typical day? Furthermore, this section states that, “To obtain a composite digital screen engagement items we converted the measures to a 10 point scale, and then took the means, deleting those adolescents who had missing values in one or more of the four items.” How/why was the 1-8 likert scale converted to a 10-point scale?
5. Time-use diaries of digital engagement: the measure section currently states that, “for those adolescents who did participate in digital screen engagement, we measured the total time spent on such activities”. For clarification, were those who did not report any digital engagement that day, were they coded as 0, or excluded from analyses using this “time spent” variable? If they were not included, how did this affect the sample size given the use of listwise deletion?

Experimental design

No further comment

Validity of the findings

No further comment

---

## Round 0.3 · accepted · Accept

Thanks so much for making these revisions. The clarity of the manuscript is greatly enhanced, and I am satisfied that you have adequately responded to each to each of the reviewer's critiques. Congratulations!